# Fast AHRS Filter for Accelerometer, Magnetometer, and Gyroscope Combination with Separated Sensor Corrections

**DOI:** 10.3390/s20143824

**Published:** 2020-07-09

**Authors:** Josef Justa, Václav Šmídl, Aleš Hamáček

**Affiliations:** 1Department of Measurement and Technology, University of West Bohemia, 30100 Plzen, Czech Republic; hamacek@ket.zcu.cz; 2RICE, University of West Bohemia, 30100 Plzen, Czech Republic; vsmidl@rice.zcu.cz

**Keywords:** attitude estimation, complementary filter, gradient descent filter, sensor fusion

## Abstract

A new predictor–corrector filter for attitude and heading reference systems (AHRS) using data from an orthogonal sensor combination of three accelerometers, three magnetometers and three gyroscopes is proposed. The filter uses the predictor—corrector structure, with prediction based on gyroscopes and independent correction steps for acceleration and magnetic field sensors. We propose two variants of the filter: (i) one using mathematical operations of special orthogonal group SO(3), that are accurate for nonlinear operations, for highest possible accuracy, and (ii) one using linearization of nonlinear operations for fast evaluation. Both approaches are quaternion-based filter realizations without redundant steps. The filters are compared to state of the art methods in this field on data recorded using low-cost microelectromechanical systems (MEMS) sensors with ground truth measured by the VICON optical system. Both filters achieved better accuracy than conventional methods at lower computational cost. The recorded data with ground truth reference and the source codes of both filters are publicly available.

## 1. Introduction

Digital attitude and heading reference systems (AHRS) are an essential component of navigation and localization systems. It is typically based on the measurements of inertial motion units consisting of gyroscopes and accelerometers in combination with magnetometers. This combination is known as the magnetic angular rate and gravity sensor combination (MARG) or magnetic inertial motion unit (MIMU). The MARG combination takes advantage of a diverse direction of gravity acceleration vector and magnetic vector field in the majority of locations on Earth (the exception being magnetic poles). The AHRS system aims to map orientation between the object of interest (body coordinate) and the coordinate system defined by the gravitational vector and the local dominant magnetic field vector.

Initial approaches to this problem were based on the analysis of instantaneous measurement. A landmark paper by Grace Wahba defined the main rotational problem for spacecraft attitude [1] in 1965. The aim is to find the right rotation matrix between two sets of vectors with the minimum weighted square error. Many follow-up contributions focused on the efficient solution of this problem, including the QUaternion ESTimator (QUEST) [2], the ESOQ algorithm, which solves the polynomial characteristic equation of QUEST in one step [3], and many others [4]. Since the first solution, the AHRS could start to grow into various fields and combine multiple sources of information. Mims and Farrell used the IMU reference for synthetic aperture radars in the 1970s [5]. In 2000, the magnetometer was used to obtain heading in a system without the gyroscope [6]. The combination of all three types (MARG) was used in 2004 by Gebre-Egziabher [7].

However, since measurements of all sensors are susceptible to disturbances, the challenge of any fusion method is to reject these disturbances as much as possible. This is often achieved using by a combination of different temporal characteristics of individual sensors. While a gyroscope provides excellent information about the rapid changes of the orientation, it provides only relative changes of orientation that are subject to slow drift varying with its lifetime and with temperature. On the other hand, an accelerometer and magnetometer provide a direct measurement of orientation; however, the measurement cannot be used as a reference at every instant since the accelerometer measures the gravity acceleration vector contaminated by dynamic acceleration and the magnetometer measures the Earth magnetic field contaminated by local magnetic anomalies and various magnetic disturbances. This contamination is also considered a disturbance and needs to be rejected. The character of disturbances may differ in different application scenarios. For example, dynamic accelerations of a MARG mounted on the foot of a running man would be different to that of a MARG mounted on an aircraft or UAV. Each sensor type is also influenced by orthogonality misalignment, cross-axis sensitivity deviation, and amplitude and phase attenuation over frequency. Our primary application is the human motion recognition domain [8,9] so we focus our attention on the specifics of this domain such as dynamic acceleration disturbance.

The majority of the fusion methods follow the predictor–corrector structure, where the gyroscope measurements are used in the prediction step and the accelerometer and magnetometer measurements in the correction step. Various classes of design methods for these filters are used, such as the Kalman filter approach, complementary filter, or gradient-based predictor–corrector filters. These approaches differ in their assumptions made on the character of the disturbances.

The Kalman filter is based on stochastic filtering theory, which assumes that the disturbance follows the Gaussian distribution of both the state evolution error and observation errors. It is a classical approach [10] used in the human motion domain with a large number of variants, see e.g., [11] for their comparison. Various extensions focus on different aspects of the filtering differing such as the effect of nonlinear models that is addressed by the more computationally costly unscented Kalman Filter [12]. However, the limiting factor of Kalman-based filters is the underlying assumption of Gaussian distributed disturbance. This has been addressed by extensions proposing adaptive estimates of the covariance matrices, see e.g., [13] for an excellent review. The approaches are based on various approaches monitoring the evolution of estimation errors using fuzzy rules [13], moving average covariance estimation [12], Markov chains [14], or segmented moving average covariance estimators [15] to name a few examples. The performance benefit of these methods usually comes with a significant computational price, making it unsuitable for low cost, low-power applications. While multiple techniques decreasing computational cost of the Kalman filters exists [16], adaptation of covariance matrices is always expensive.

The complementary filtering [17,18,19] is using low- and/or high-pass filters to remove disturbances from the measurements. The disturbance is thus modeled as any signal that does not pass through the filter. These filters are computationally much cheaper than Kalman filters, see e.g., [20]. The tuning of those filters is still complicated, especially in time-varying scenarios where the character of disturbances changes.

The third principal direction of handling the disturbance is the gradient-based approach pioneered by Madgwick [21], which is based on the assumption that valid measurement is only within a predefined interval and anything beyond is a disturbance. While other gradient-based approaches exist, using different update rules such as Gauss–Newton [22], the approach of [21] is unique in the recommendation to normalize the gradient, which is a computationally cheap operation. Similar ideas are known in machine learning as gradient clipping [23]. Its excellent empirical performance was recently explained using the assumption of relaxed Lipschitz constant of the optimized cost function [24].

In this paper, we follow the approach presented in [21], since it corresponds to our experience that both accelerometer and magnetometer suffer from severe disturbances. Since these disturbances do not follow any predefined probability distribution, neither frequency spectrum, the limitation of their amplitude (i.e., limit on the Lipschitz constant of the dynamical model) seems a reasonable assumption to make. However, in [21] this assumption is applied to the difference between gyroscope prediction and fused measurement vector that is obtained using the solution of the Wahba problem. We conjecture that this is suboptimal since the amplitude of the disturbance affects the direction of the correction step. Therefore, we designed a new correction step that puts a hard limit on the amplitude of the disturbance of individual sensors. Similar ideas on the separation of the sensor influence have been proposed for the Kalman filtering approach in [25,26], with a different assumption on the noise. It is also related to various two-step approaches [27], however, in a more principled manner.

The contribution of our paper is as follows:We propose a new gradient-based filter for AHRS with the following features: (i) the gradient of correction from magnetometer and accelerometer are processed independently, (ii) the step size of the gradient descent is limited by the correction function independently for each sensor, and (iii) the correction vectors are fused using a new approximation of the correct SO(3) operation. Separation of the correction steps from each sensor implied heavy modification of the original filter, as visualized in Figure 1.For each operation that is used in the design of the proposed filter, we analyzed the accuracy of its implementation in the SO(3) group. We provide two approximations: (i) an accurate approximation of the correct quaternion operation as possible, and (ii) its approximation simplified using e.g., linearization. The proposed filter is presented in two versions: a separated correction filter (SCF) using the accurate SO(3) operations, and a fast SCF (FSCF) that is composed of the simplified (linearized) operations.We recorded measurements of the MARG sensor mounted on an experimental platform in a laboratory equipped with the optical tracking system VICON-460. We provide this data as an open-source benchmark. We compare the performance of the proposed filter with selected alternative filters and demonstrate that it has higher accuracy at a lower computational cost.

The remainder of this paper is organized as follows: Elementary operations with quaternions and their notation is reviewed in Section 2. The proposed filter is described in detail in Section 3 including a description of several possible approximations of the correct SO(3) operations. Experimental results are provided in Section 4, including details of the data measurement and comparison of selected competing filters. A conclusion from the experiments is drawn in Section 5.

## 2. Important Rotation Operations

In this section, the necessary rotational operations that are used in the designed filters are reviewed. For a full treatment of the subject, see [28]. First, rotation of normalized vectors in Euclidean space is presented, then elementary operations with quaternions are described. Unit vector *a* is a vector that has unit norm $\left||a|\right|=a^{⊤}a=1$, where ${}^{⊤}$ denotes transposition. The direct angle (DA), $\alpha_{a,b}$, between two unit vectors, $a={\left[a_{x},a_{y},a_{z}\right]}^{⊤},$ and $b={\left[b_{x},b_{y},b_{z}\right]}^{⊤}$, is defined by
(1)$$\begin{array}{cc} \alpha_{a,b} & =\cos^{-1}\left(a^{⊤}b\right). \end{array}$$

The cross product of two unit vectors *a* and *b*, is defined by
(2)$$\mathrm{cross}\left(a,b\right)=\left[a_{y}b_{z}-a_{z}b_{y},a_{z}b_{x}-a_{x}b_{z},a_{x}b_{y}-a_{y}b_{x}\right].$$

The proposed filter is based on Hamilton’s quaternions. For implementation in this mathematical group, we use a unit quaternion which is a complex number representing rotation from the source coordinate frame *f* to the destination frame $f^{\prime }$ which can be written in vector form
(3)$$q_{f,f^{\prime }}=\left[q_{w},q_{x},q_{y},q_{z}\right]=\left[\cos\frac{\theta }{2},u_{x}i\sin\frac{\theta }{2},u_{y}j\sin\frac{\theta }{2},u_{z}k\sin\frac{\theta }{2}\right],$$
where unit vector *u* in coordinate frame *f* defines rotational axis (axis around rotation has been done) and θ the angle of rotation. The angle of rotation θ corresponds to the DA for 3D vectors.

The quaternion rotation *q* can be converted to the rotation matrix *R* using
(4)$$R=\left[\begin{array}{ccc} q_{w}^{2}+q_{x}^{2}-q_{y}^{2}-q_{z}^{2} & 2q_{x}q_{y}-2q_{w}q_{z} & 2q_{x}q_{z}+2q_{w}q_{y}\\ 2q_{x}q_{y}+2q_{w}q_{z} & \;\;\;q_{w}^{2}-q_{x}^{2}+q_{y}^{2}-q_{z}^{2}\;\;\; & 2q_{y}q_{z}-2q_{w}q_{x}\\ 2q_{x}q_{z}-2q_{w}q_{y} & 2q_{y}q_{z}+2q_{w}q_{x} & q_{w}^{2}-q_{x}^{2}-q_{y}^{2}+q_{z}^{2} \end{array}\right]$$

The most important operations with quaternions are the quaternion product × and quaternion conjugation $q^{-1}$, since they define the rotation of coordinates of vector *v* in frame *f* to coordinates $v^{\prime }$ in frame $f^{\prime }$
(5)$$\begin{array}{cccc} v^{\prime } & =q_{f,f^{\prime }}^{-1}\times v\times q_{f,f^{\prime }}. & q_{a,c}^{-1} & =\left[q_{w},-q_{x},-q_{y},-q_{z}\right]=q_{c,a}. \end{array}$$

Here, *v* is a four-dimensional vector with the first element equal to zero followed by conventional coordinates in 3D space. The quaternion multiplication ×,
(6)$$q_{f,f^{\prime \prime }}=q_{f,f^{\prime }}\times q_{f^{\prime },f^{\prime \prime }},$$
is the result of the product of complex numbers corresponding to vectors (Equation 3). Since $q_{w}=\cos\frac{\theta }{2}$ and $\sqrt{q_{x}^{2}+q_{y}^{2}+q_{z}^{2}}=\sin\frac{\theta }{2}$ (Equation 3), the angle of rotation θ can be computed from a quaternion using:(7)$$\begin{array}{cc} \theta (q) & =2\mathrm{arctg}2(\sqrt{q_{x}^{2}+q_{y}^{2}+q_{z}^{2}},q_{w}). \end{array}$$

## 3. Filter Implementation

The filter is designed as a modification of the Madgwick filter where we split the contribution of each sensor to the correction. This requires a replacement of the solution of the Wahba problem. We thus cannot fuse information in the derivative form as the original filter, and we have to use the absolute form. Specifically, the correction step is composed of two important parts for each sensor independently: (i) determination of distance and direction of deviation from the prediction, and (ii) determination of the correction step size based on the shape of a correction function. The correction rotation is determined using a combination of the directions of deviation of individual sensors weighted by their correction step sizes.

The proposed approach has an independent weighting of their contributions to correction. Note that the correction parts from the accelerometer and magnetometer are almost completely separated except the magnetic reference vector determination (more in Section 3.3). There were some attempts to do the separation of sensor contributions in previous works, such as [26], with which we compare our filter in the experimental part.

The proposed filter can be implemented using either quaternions or rotational matrices. We present the majority of operations in the quaternion formulation and use the rotational matrix in cases where it is computationally advantageous. Due to limited computing power in embedded systems (the most common application), we present two versions of the filter. The separated correction filter (SCF) using as accurate numerical operations, and the fast separated correction filter (FSCF) using computationally efficient approximations of these operations. The influence of those approximations was studied in simulations. The structure of those two variants is identical, the difference is only in details of the implementation of individual blocks (Figure 1 right). The output of the filter is quaternion ${\hat{q}}_{t}$ defining rotation from the estimated coordinate frame of the sensor body to the earth coordinate frame. Here, *t* is used as the index of the time step. Implementation of each block from Figure 1 (right) is described as follows.

### 3.1. Dynamic Rotation Change (qDot)

Transformation of the gyroscopic rotation rate data to the small rotation difference is provided by the dynamic rotation block (qDot). The function can also be interpreted as a transformation of the gyroscopic measurement to the rotational quaternion. The first step is the transformation of the gyroscopic data to Euler angles (yaw, pitch, and roll, respectively) in the body coordinate frame:(8)$$\left[\begin{array}{c} \alpha \\ \beta \\ \gamma \end{array}\right]=\omega_{t}\;dt,$$
where dt is the sampling rate. The second step is the transformation of the Euler angles to the quaternion by
(9)$$q_{dot,x}=\sin\left(\frac{\alpha }{2}\right),\;q_{dot,y}=\sin\left(\frac{\beta }{2}\right),\;q_{dot,z}=\sin\left(\frac{\gamma }{2}\right),$$
(10)$$q_{dot,w}=\sqrt{1-\left(q_{dot,\;x}^{2}+q_{dot,\;y}^{2}+q_{dot,\;z}^{2}\right)},$$
where $q_{dot}$ is the rotation between coordinate frame of the body at time t−1 and time *t*.

The Equations (Equation 9) and (Equation 10) are used in SCF. Due to small angles (fast sampling rate) it is possible to linearize the sine functions in (Equation 9) for the use in FSCF
(11)$${\tilde{q}}_{dot}=1+\frac{\alpha }{2}i+\frac{\beta }{2}j+\frac{\gamma }{2}k.$$

The impact of this approximation is negligible in real conditions.

### 3.2. Prediction Integration (qPred)

Prediction of the new position reference frame of the sensor based on information from the gyroscope is obtained by rotation integration in SO(3) which is defined by quaternion multiplication (Equation 6),
(12)$$\begin{array}{cc} q_{pred} & ={\hat{q}}_{t-1}\times q_{dot}, \end{array}$$
where $q_{pred}$ is a quaternion of rotation from the predicted frame of the body to the earth coordinate frame.

### 3.3. Accelerometer Reference Vector (aRef, accPred)

We assume that the measured vectors of acceleration are normalized at the beginning of each filter step (Sensors have to be calibrated before measurement, especially the magnetometer). By the reference vector of the accelerometer, we define the part of the earth coordinate frame. The reference vector can be set as
(13)$$a_{r}={\left[0,0,1\right]}^{⊤},$$
which represents zero rotation if the gravitational acceleration is pointing down along the *‘z’* axis of the body reference frame.

The reference vector in the predicted body frame can be obtained by quaternion rotation via Equation (Equation 5)
(14)$$a_{pred}=q_{pred}^{-1}\times a_{r}\times q_{pred}.$$

However, for the specific choice of the reference vector (Equation 13), the same operation using rotation matrices has a lower computational cost. Therefore, we implement (Equation 14) in block “accPred” using
(15)$$a_{pred}=R^{⊤}a_{r}=\left[\begin{array}{ccc} 0 & 0 & 2q_{x}q_{z}-2q_{w}q_{y}\\ 0 & 0 & 2q_{y}q_{z}+2q_{w}q_{x}\\ 0 & 0 & 1-2\left(q_{x}^{2}+q_{y}^{2}\right) \end{array}\right]\left[\begin{array}{c} 0\\ 0\\ 1 \end{array}\right].$$

### 3.4. Magnetometer Reference Vector (mRef, magPred)

The choice of the reference vector for the magnetometer is more complicated. Due to magnetic distortions, influences by ferromagnetic materials in buildings and variation of magnetic inclination, it is not appropriate to define the magnetic reference vector by a constant.

We follow [21], and choose the reference of the magnetometer using the predicted reference frame. The original reference proposed in [21] comes with a significant computational cost. Therefore, we propose to compute the reference in the body frame instead of the Earth frame. To achieve this, we need to rotate the accelerometer to the body frame. We use the predicted value of the accelerometer for suppression of the measurement noise. The result is a computationally cheaper alternative since its components are already available:(16)$$\begin{array}{cc} m_{r,z}= & \cos\left(\hat{\varphi }\right)=a_{pred,x}m_{t,x}+a_{pred,y}m_{t,y}+a_{pred,z}m_{t,z},\\ m_{r,x}= & \sin\left(\hat{\varphi }\right)=\sqrt{1-m_{r,z}^{2}},\\ m_{r}= & {\left[m_{r,x},0,m_{r,z}\right]}^{⊤}, \end{array}$$
where $\hat{\varphi }$ is the estimate of the current inclination between the predicted accelerometer and magnetometer measurement using (Equation 1). This formulation is computationally faster and the experiments show that it has a negligible impact on the accuracy of the resulting filter.

Rotation of the reference vector to the predicted coordinate frame is obtained analogically to (Equation 15) as
(17)$$m_{pred}=R^{⊤}m_{r}=\left[\begin{array}{ccc} 1-2\left(q_{y}^{2}+q_{z}^{2}\right) & 0 & 2q_{x}q_{z}-2q_{w}q_{y}\\ 2q_{x}q_{y}-2q_{w}q_{z} & 0 & 2q_{y}q_{z}+2q_{w}q_{x}\\ 2q_{x}q_{z}+2q_{w}q_{y} & 0 & 1-2\left(q_{x}^{2}+q_{y}^{2}\right) \end{array}\right]\;\;\left[\begin{array}{c} m_{r,x}\\ 0\\ m_{r,z} \end{array}\right].$$

### 3.5. Deviation of Prediction and Measurement (magDev, accDev)

If the predicted reference frame is correct, the predicted values of the magnetometer and accelerometer reference vectors should be close to the measured values. We quantify the deviation of the measured vectors from the reference vectors by angle and direction. For example, for the accelerometer, the deviation between the measured vector $a_{t}$ and the predicted vector $a_{pred}$ is computed as
(18)$$\begin{array}{cccc} \alpha_{a}= & \cos^{-1}\left(a_{t}^{⊤}a_{pred}\right), & a_{cor}= & \frac{\mathrm{cross}\left(a_{t},a_{pred}\right)}{∥\mathrm{cross}\left(a_{t},a_{pred}\right)∥}, \end{array}$$
where cross is defined in (Equation 2). The same operations are performed for measurements of $m_{t}$ and reference vectors $m_{r}$ of the magnetometer, yielding angle $\alpha_{m}$ and $m_{cor}$. These operations are implemented by blocks “accDev” and “magDev” in Figure 1.

### 3.6. Correction Step Determination (magCor, accCor)

Deviation of the measurement from the prediction is an indicator in which direction we should rotate the predicted coordinate frame to obtain a better estimate. The key component of the filter is the choice of the influence of the deviation on the correction of the predicted coordinate frame. In our filter, we propose a correction in the direction of vectors $a_{cor}$ and $m_{cor}$ but decrease the scale of the correction (angles $\alpha_{a}$ and $\alpha_{m}$) by a correction function, e.g.,
(19)$$\beta_{a}=f_{cor}\left(\alpha_{a},\lambda_{a}\right),$$
for the accelerometer deviation angle. Here, $\lambda_{a}$ denotes a tuning parameter. The choice of function $f_{cor}$ has a significant impact on the result. We will test three basic correction functions: (i) the linear function $\beta_{a}=\lambda_{a}\alpha_{a}$ (intercept zero); (ii) the constant function $\beta_{a}=\lambda_{a}$ (Madgwick approach); and (iii) the segmented function as a combination two previous: $\beta_{a}=\lambda_{a,1}\alpha_{a},$ if $\lambda_{a,1}\alpha_{a}<\lambda_{a,2}$, $\beta_{a}=\lambda_{a,2}$ otherwise. The most commonly used method in other filters is the linear correction step. Parameters $\lambda_{a}$ and $\lambda_{m}$ can be considered as the filter correction weights of each sensor.

Correction rotation from the accelerometer measurement can be formulated in the form of a quaternion (Equation 3) as follows:(20)$$q_{a,cor}=\left[\cos\frac{\beta_{a}}{2},a_{cor}^{⊤}\;\sin\frac{\beta_{a}}{2}\right].$$

Since angle $\beta_{a}$ is typically very small, we can approximate (Equation 20) by linearization
(21)$${\tilde{q}}_{a,cor}=\left[1,a_{cor}^{⊤}\frac{\beta_{a}}{2}\right],$$
which is used in the simplified FSCF. Analogical equations are used for the rotation towards the measurement of the magnetometer, $q_{m,cor}$.

It is not obvious, but in his famous paper [21], Madgwick presents two filters with different behavior. In his paper, the first filter ends by Equation (Equation 23). The correction propagated through the gradient filter leads to a classic linear correction function. This filter is denoted Linear Madgwick. The second filter in his paper is based on fusion by (29). The constant correction step was used in this field for the first time. We call the second filter the Madgwick gradient filter. The comparison of the Linear Madgwick filter with its final version is discussed in the experimental part, Section 4.3.1.

The correction function is implemented in blocks “accCor” and “magCor” in Figure 1. If we use the constant correction step, the deviation angle is not included in the filter step. We can then save a small amount of computing power by omitting the computation of $\alpha_{a}$ and $\alpha_{m}$ in Equation (Equation 18) from the previous block.

It is clear that for a very small angle deviation, the constant correction step will lead to overshooting and thus to the zig–zag effect. This can be prevented by the use of the segmented function. A comparison of the effect of the tested correction functions is provided in the experimental part, see Section 4.3.2.

### 3.7. Fusion of Correction Quaternions (fuseCor)

The fusion step aims to create a single quaternion that represents a correction of the predicted reference frame towards the measurements. In Euclidean space, this is often achieved by convex combination. An equivalent operation in rotational space can be obtained numerically, by repetitive application of infinitesimal steps of alternating rotations toward the first and the second measurement. Since this is a computationally expensive operation, the most common approximation of this operation is to apply rotational multiplication of fused quaternions [21], which corresponds to a coarse approximation of the correct step. However, this simple approximation does not preserve commutativity and its accuracy is quickly decreasing with increasing angles and difference in direction of the fused quaternions.

As an alternative, we propose a simple analytical formula that approximates the numerical solution. We propose a three-step approximation. First, create a convex combination of the vectors $a_{cor}$ and $m_{cor}$
(22)$$f_{cor}=\left(\frac{\beta_{a}}{2}a_{cor}+\frac{\beta_{m}}{2}m_{cor}\right),$$
using weights $\beta_{a}$ and $\beta_{m}$. Second, compute its curvature on the hypersphere
(23)$$c=\sin c\left(\mathrm{norm}\left(f_{cor}\right)\right),$$
where sinc(x)=sin(x)/x is the unnormalized sinc function. Third, generate quaternion
(24)$$\begin{array}{cc} q_{cor} & =\left[\sqrt{1-f_{cor}^{⊤}f_{cor}c^{2}},f_{cor}c\right], \end{array}$$
which is the final correction quaternion. For small rotations, the curvature *c* is approaching one, which allows simplifying (Equation 24) for the FSCF:(25)$$\begin{array}{cc} {\tilde{q}}_{cor} & =\left[1,f_{cor}\right], \end{array}$$

The accuracy of the proposed formula was tested in comparison with a numerical solution using a billion steps for each testing rotation. The accuracy of the tested methods for two rotations of perpendicular directions with the same value of the rotation angle is displayed in Figure 2 for increasing value of the rotation angle.

Note that the proposed Formula (Equation 24) is accurate for the whole tested interval of the rotation angles. The simplified formula is accurate for low values of the rotation angle but deteriorates over 30 degrees. The commonly used method of quaternion multiplication is the least accurate in this case.

### 3.8. Application of Correction Step (Correct)

The final filter block “correct” applies the correction step to the prediction by quaternion multiplication as
(26)$${\hat{q}}_{t}=q_{pred}\times q_{cor}.$$

The FSCF filter contains few approximations of the mathematically correct SO(3) operations, (Equation 11), (Equation 21), and (Equation 25). The normalization of the estimated quaternion has to be done after each step of the filter. This is highly recommended also for SCF because of the numeric error integration.

## 4. Experimental Results

A comparison of the accuracy of AHRS filters is a problematic task since each filter has different behavior in different situations. Therefore, we prepared an experimental platform and recorded three different datasets with different characteristics.

We evaluated the performance of the filters in two steps. First, we calculated the rotation between the ground truth quaternion and the estimated quaternion
(27)$$\begin{array}{cc} q_{dif} & =q_{gt}\times q_{est}^{-1}. \end{array}$$

Second, we evaluated the rotation angle of the difference $\theta (q_{dif})$ using (Equation 7) in each time step of the experiment. Accuracy of the estimation was evaluated using two common measures, the mean absolute error (MAE) and the root mean square error (RMSE) of the rotation angle $\theta (q_{dif})$. We used MAE for graphical presentations since it is distinguishable for the large error range.

To minimize the potential of introducing an error in implementation, we preferred to compare the quality of our filter with the methods published with implemented code. Therefore, we chose the Madgwick’s complementary filter [21], the Valenti’s complementary filter [18], and the Guo’s Fast Kalman Filter [16]. However, none of the recently published filters with independent corrections from the magnetometer and accelerometer had available code. Therefore, we compared the methods with our implementation of Suh’s filter [26].

### 4.1. Data Acquisition

The experimental platform consisted of three perpendicular rods of lengths 36 cm to obtain better localization accuracy (Figure 3). The platform was equipped with low-cost sensor unit BMX055 from BOSCH Sensortec and nrf24l1s unit configured to act as a wireless transceiver. The data were captured by the second nrf24l1s unit operated as a wireless receiver connected to the Arduino UNO where they were mixed and logged with data coming from the VICON-460 optical positioning system. Accuracy of the positioning was studied in [29,30] confirming sub-millimeter accuracy of the position measurements, depending on operating conditions. Since even the worst-case error of 1 mm on the position of the rod implies a 0.16 degree error on its rotation, we considered this measurement to be the ground truth reference for the tested filters.

The recorded datasets and implementation of the SCF and FSCF are freely available (https://github.com/Josef4Sci/AHRS_Filter).

Three experiments were performed in the laboratory to capture different scenarios that can occur in practical applications. The first experiment was designed to contain relatively slow rotations, whereas the second experiment was designed to contain fast rotations. In the third experiment, the rotations are inhibited but the platform was subject to dynamic accelerations in different directions. Datasets recorded in each experiment were synthetically connected to create three different scenarios. The first scenario represents favorable conditions where only slow rotations are present, therefore only the dataset from the first experiment is used. This scenario is common e.g., in indoor drone applications. The second scenario represents more demanding conditions where both slow and fast rotations are common, as arise e.g., in virtual reality applications. This scenario is obtained by joining datasets from the first two experiments. The third scenario is the harshest conditions where rotations are disturbed by abrupt accelerations from arbitrary directions, as arise in inertial motion capture systems. This is obtained by joining data from all three experiments. Data from all three datasets are displayed in Figure 4.

### 4.2. Tuning of Filter Parameters

The tuning of filter parameters is essential for proper evaluation of their performance [31]. Therefore, we optimized the tuning parameters of all filters for each scenario and each performance measure (MAE, RMSE) independently. Since the problem is nonconvex, we needed a global optimization method. For this reason, many authors use a stochastic search, such as the particle swarm optimization [32], which may not be easily reproducible. Therefore, we investigated deterministic optimization methods. The conventional Nelder–Mead simplex search as implemented in Matlab was found to be unreliable. Therefore, we designed our version of the refined grid search [33]. Specifically, we started with an initial guess and designed a grid of values consisting of values of the initial guess multiplied by predefined coefficients. See supplementary material for details of implementation.

### 4.3. Influence of the Correction Function

The idea of the correction in its basic form was proposed in [21] and elaborated in a more advanced form in the proposed method. It was found to be a very important factor for the accuracy of different filters as is demonstrated in this Section. To obtain fair testing conditions, parameters of all tested filters were optimized for each test and each filter to get the best possible response. The errors of the filters are quite noisy in raw format, therefore we present them processed by the moving average filter with a window size of 40 samples for better readability (sampling rate about 100 Hz).

#### 4.3.1. Correction Function in Madgwick Filters

The basic filter presented in [21] is using the linear function and will be denoted as Linear Madgwick. However, the main results presented in [21] are using the constant correction function and will be denoted as Madgwick in our comparison. The difference between those two versions is demonstrated in Figure 5 on the data from the first scenario. The version with constant correction function outperforms the linear filter by 58%. From now on, only the filter with constant correction function version will be used for tests.

#### 4.3.2. Correction Function in SCF

Three shapes of the correction function proposed in Section 3.6 were tested: (i) the linear function, which has only one parameter per correction sensor; (ii) the constant function with one parameter per sensor as well; (iii) the segmented function with two parameters per correction sensor. To reduce the number of parameters in the segmented function, we tuned a single gain of the linear part for both sensors. This leads to a filter with three parameters for tuning.

The parameters of all correction functions were optimized for performance on scenario 3 and the results are displayed in Figure 6. The result shows that there is almost no difference in the accuracy of the filter with the constant correction function and its theoretically improved version, the segmented function. Note that the error of the filter with linear correction function grows significantly between 23 and 36 s (the part of heading change, Figure 6). Another major difference is also in acceleration disturbances. The influence of the correction function in fast rotations seems to be insignificant. In this part, the error is dominated by a peak between 60 and 70 s, which could be caused by the desynchronization between the reference and measured data.

The average error is summarized in Table 1.

Note that the impact of the segmented function on the error is low but tuning of the additional parameter is hard. Therefore, the constant correction function was used in the following tests.

### 4.4. Comparison on Selected Application Scenarios

The performance of the tested filters was evaluated for all three scenarios defined in Section 4.1. Parameters of all filters were tuned for each scenario independently to obtain the lowest average MAE or RMSE. The parameters are summarized in Table A1 in Appendix A. Since the SCF and FSCF have almost the same behavior, only the FSCF results are displayed in Figure 7, Figure 8 and Figure 9. Since the results of Suh’s filter [26] were significantly worse than other filters in the testing set, its estimates are displayed as transparent for better readability of the results of other filters. Results for the first scenario with only slow rotation are displayed in Figure 7 and summarized in Table 2. The performance of all filters is comparable in the first half of the dataset but differs in the second where the proposed filter has the lowest error.

The second scenario contains both slow and fast rotations. The results are summarized in Table 2 and displayed in detail in Figure 8. Note that the error during fast rotations (40–70 s) is much higher than that for the slow rotations and is dominated by a peak between 60–70 s. This peak can be also explained by deviation in the reference VICON data. However, since all filters have almost the same performance in this region, it has no impact on the results of the comparison. The filters were tuned for the best overall results in this scenario but the performance in the first part (slow rotations) is similar to results with parameters tuned only for the first scenario.

The last scenario complements slow and fast rotations by a sequence of dynamic accelerations to test robustness to disturbance. Once again, all filters were tuned for best overall performance in the whole data set. The results are summarized in Table 2 and analyzed in detail in Figure 9. Note that the proposed filter is significantly better than the competitors in the second part of the slow rotation data and in the dynamic acceleration data. In all other cases, it is comparable to the best of the remaining filters.

### 4.5. Cross-Validation Study

The previous experiments were optimized for best performance to obtain theoretical lower bound. In this section, we present the results of our investigation of the sensitivity of all tested methods to the tuning of their parameters. We performed a cross-validation study, in which we trained parameters of the methods on one part of the data and report their performance on the remaining data. Specifically, the introduced dataset is composed of the distinct parts: (i) slow rotation 0–40 s, that form scenario 1, (ii) fast rotations 40–72 s, and (iii) dynamic accelerations after 72 s. Since tuning of the parameter on the dynamic acceleration would provide degenerate behavior (the correct output is always zero), we optimize parameters of all methods only on one of the fast or slow parts and evaluate their performance on the remaining part of the data set, Table 3. The best performance is achieved for the proposed method, followed by the original Madgwick filter. This is an advantageous property of the gradient clipping that is insensitive to the absolute value of the error.

### 4.6. Computational Costs

The computational cost is an important factor for embedded systems containing MARG sensor combination. Processing a real-time data flow is the most common task for AHRS filters. The speed of filters can be compared via the number of floating-point operations in one filter step. The results for all tested methods are listed in Table 4.

The computational time of the Madgwick-type filters can be further reduced using implementation tricks techniques of [8] or the simplification of the problem [34]. While [8] achieves almost identical accuracy as [21], the simplification presented in [34] has a negative impact on accuracy, in our case 57% increase in MAE over [21] in scenario 3.

We propose a simplification of the magnetic reference in Section 3.4. This simplification allowed for the reduction of the computational cost of SCF and FSCS with negligible impact on accuracy. Replacing the proposed magnetic reference by the original reference of [21] lead to a 0.1% improvement of MAE on Scenario 3 of our experiments.

### 4.7. Summary of the Experimental Part

The benefit of using correct mathematical operations in SO(3) was found to have a negligible impact on the filter performance. Specifically, the improvement of the proposed method for quaternion fusion over the quaternion multiplication (Figure 2) was 0.3‰ for the method ignoring rotational curvature and 0.4‰ for the method with inverse curvature. However, it may become more important in applications with lower sampling frequency.

A more important factor in filter performance is the choice of the correction function, as studied in Section 4.3.2. The constant correction step was found to be a good choice for standard applications with MARG sensor combination. Methods that use the correction function (such as the constant correction step) are also less sensitive to tuning parameters and perform well in cross-validation study.

However, the main contribution to the good performance of the proposed filter is the application of the correction function independently to data from accelerometer and magnetometer before their fusion. In our tests, the optimal weight of the magnetometer was approximately 10 times smaller than the weight of the accelerometer. This indicates that filters with common weight for both accelerometer and magnetometer are sub-optimal in real applications.

The proposed filters were found to be more accurate than any of its competitors in all tests. Specifically, the mean absolute error of the proposed filter was 77% of the second-best filter in scenario 1 (slow rotations), 93% of the second-best in scenario 2 (slow and fast rotations), and 86% in scenario 3 (slow and fast rotations, and dynamic acceleration).

The biggest advantage of the proposed filter is in its accuracy for slower rotations and dynamic accelerations. Nevertheless, the improvement for faster rotations is also significant.

## 5. Conclusions

A novel predictor—corrector AHRS filter using separate correction steps for magnetometer and accelerometer (SCF) and its faster version (FSCF) is proposed. Both filters have lower computational cost than other well-known methods in this field. Although both filters have simple structures, it was demonstrated that they are more accurate than their competitors. The most visible improvement was on a scenario with slow rotations (77% mean absolute error of the second-best filter) but it improves on the dynamic acceleration as well as fast rotations. The key difference from previous approaches is the application of correction function independently to data from accelerometer and magnetometer before their fusing. This opens possibilities for future research, e.g., on-line adaptation of the weights. We also provide our testing dataset for open access for better reproducibility of our results.

## Figures and Tables

**Figure 1 sensors-20-03824-f001:**
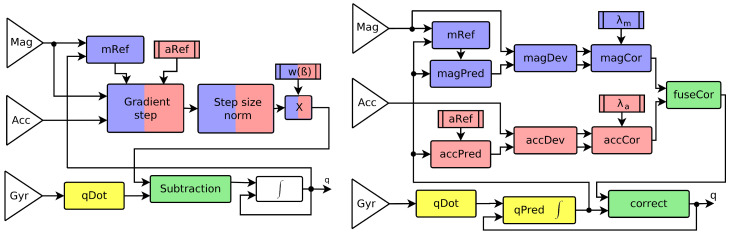
Block diagram of the Madgwick magnetic angular rate and gravity sensor combination (MARG) filter (**left**) and the proposed separated correction filter (SCF) (**right**). **Yellow** blocks denote prediction from the gyroscope; **Blue** blocks denote correction from the magnetometer; **Red** blocks the accelerometer correction, and **Green** is fusion of the correction steps and its application to the predicted values. Detailed description of blocks of the proposed filter is provided in Section 3, namely Section 3.1 (qDot), Section 3.2 (qPred), Section 3.3 (aRef, accPred), Section 3.4 (mRef, magPred), Section 3.5 (magDev, accDev), Section 3.6 (magCor, accCor), Section 3.7 (fuseCor), and Section 3.8 (correct). Tuning parameters of the proposed filter are $\lambda_{m}$ and $\lambda_{a}$.

**Figure 2 sensors-20-03824-f002:**
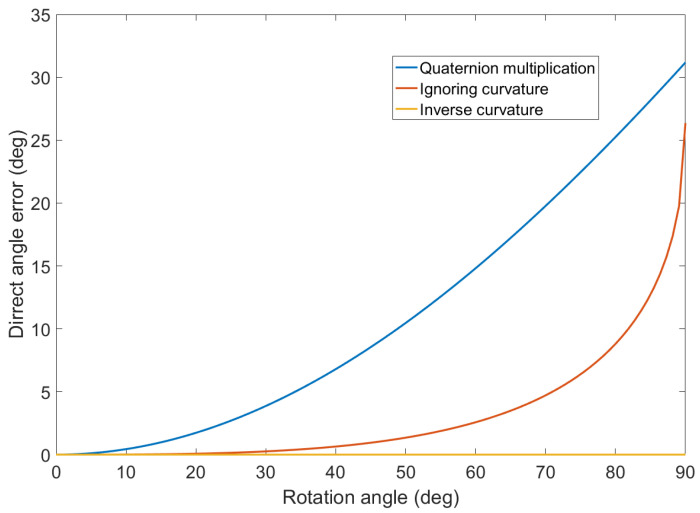
Accuracy of quaternion fusion of two rotations of perpendicular directions with the same value of the rotation angle for the quaternion multiplication (blue), the proposed method with inverse quadrature (Equation 24) (yellow), and the simplified method (Equation 25) ignoring the curvature (red). The error is computed as a deviation from the numerical solution with billion steps.

**Figure 3 sensors-20-03824-f003:**
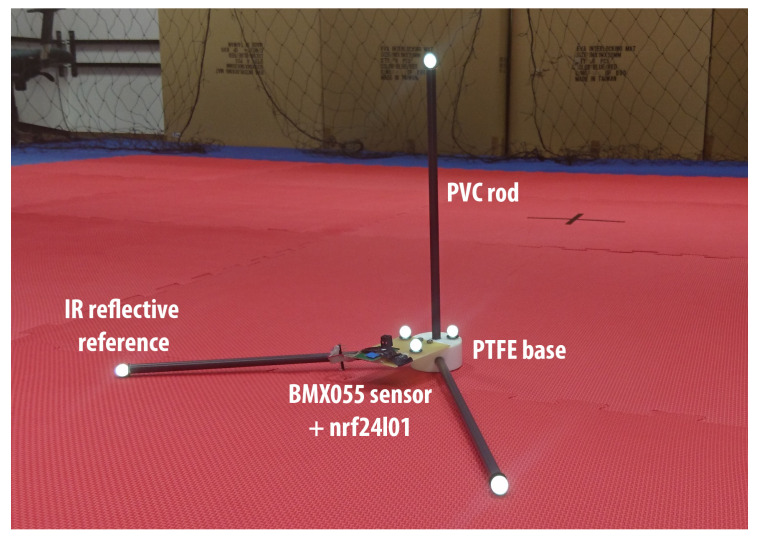
The measuring platform in the VICON laboratory.

**Figure 4 sensors-20-03824-f004:**
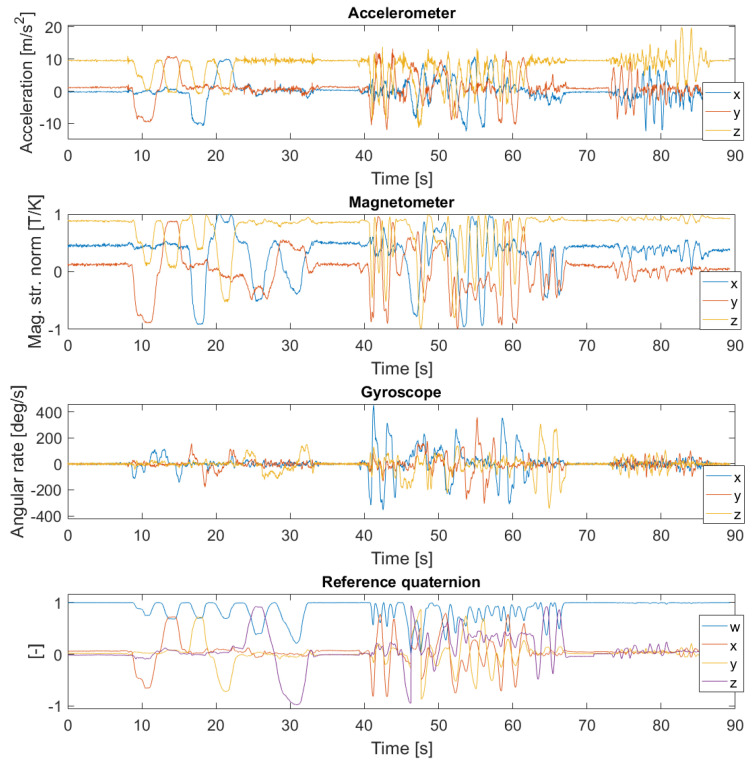
The measured data from all three experiments. The first three rows display data from the MARG sensor: all three axes, x,y,z, of the accelerometer, magnetometer, and gyroscope, respectively. The measured positions of the points on the platform from the VICON system were converted to the reference quaternion that is displayed via its vector components, w,x,y,z (Equation 3) in the bottom row. The slow rotations occur between 0–40 s, fast rotations between 40–72 s, and dynamic accelerations after 72 s.

**Figure 5 sensors-20-03824-f005:**
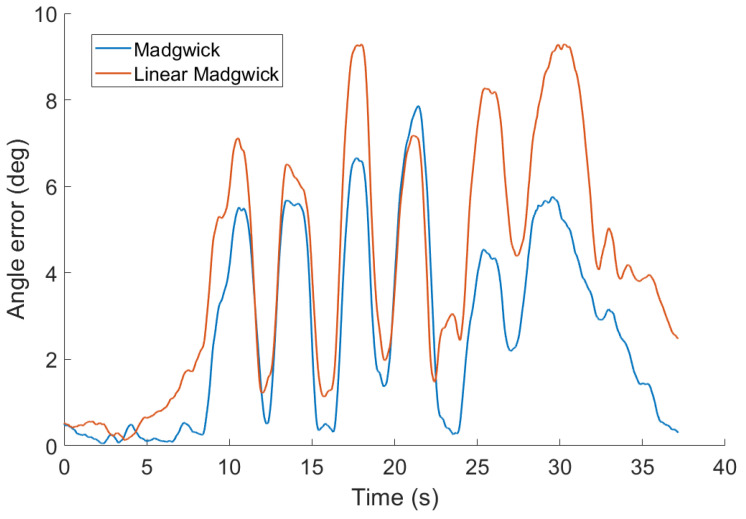
Comparison of accuracy of Madgwick’s filters for two choices of the correction function on the data from the first scenario. Parameters of all filters were tuned for the best overall performance.

**Figure 6 sensors-20-03824-f006:**
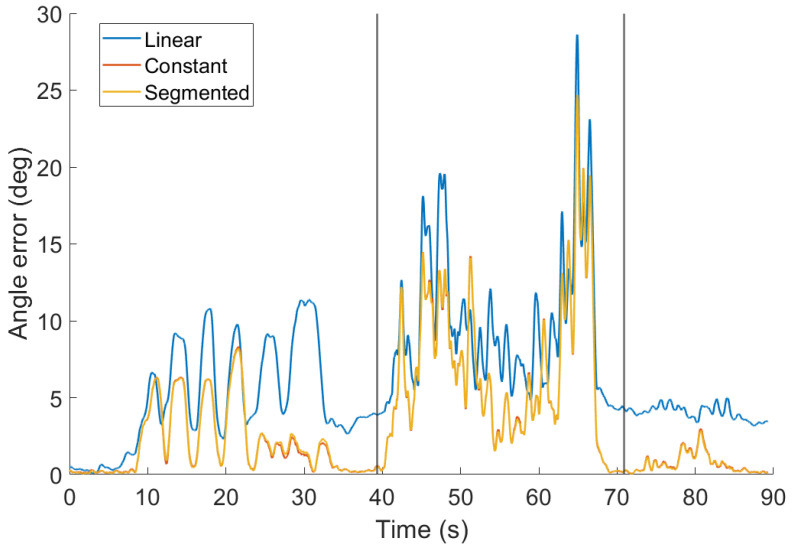
Accuracy of the separated correction filter (SCF) for scenario 3 and three different choices of the correction function: Linear, constant, and segmented. Parameters of all filters were tuned for the best overall performance.

**Figure 7 sensors-20-03824-f007:**
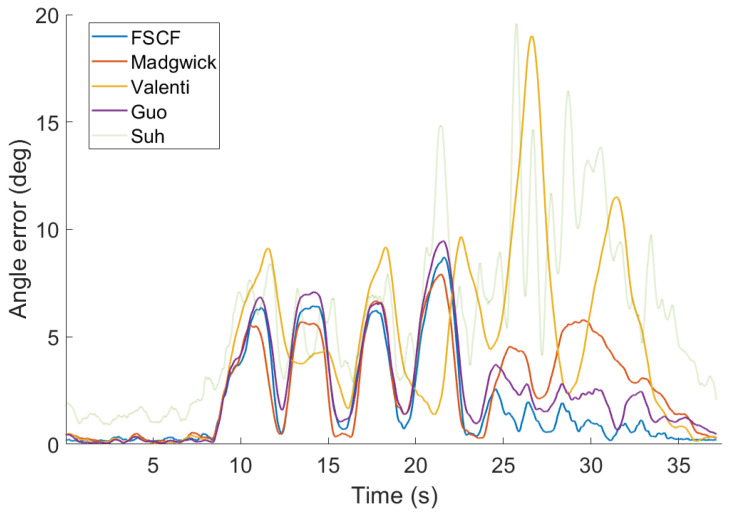
Error of the tested filters for data from the first scenario. Parameters of all filters were tuned for best overall (scenario 3) performance.

**Figure 8 sensors-20-03824-f008:**
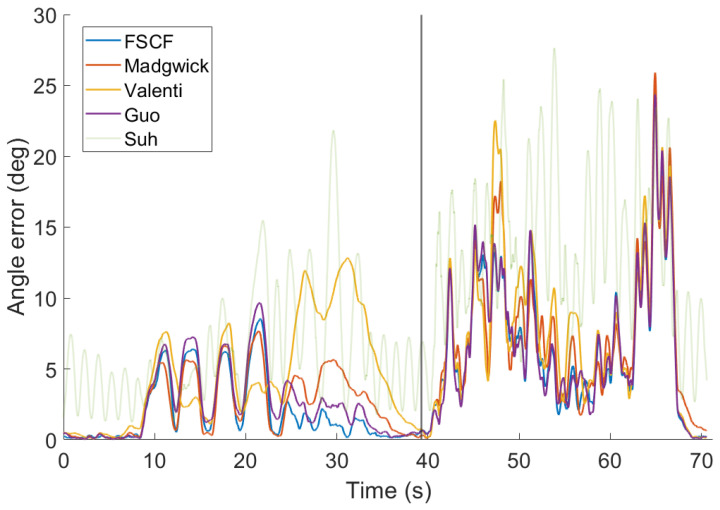
Error of tested filters for data from the second scenario. Parameters of all filters were tuned for best overall (scenario 3) performance.

**Figure 9 sensors-20-03824-f009:**
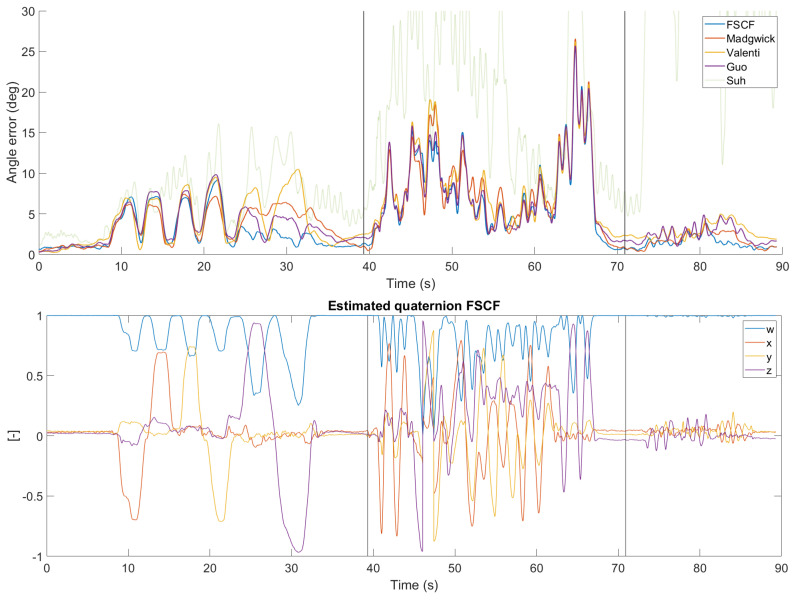
Error of all tested filters for the data from the third scenario (**top row**): slow rotations (**left**), fast rotations (middle), and dynamic acceleration (**right**). Parameters of all filters were tuned for the best overall performance. Quaternion estimated by the fast separated correction filter (FSCF) is displayed for illustration (**bottom row**).

**Table 1 sensors-20-03824-t001:** Comparison of the proposed filter with different correction function shapes.

Shape	MAE Error (Deg)
Linear	5.19
Constant	**3.37**
Segmented	**3.37**

**Table 2 sensors-20-03824-t002:** Estimation errors of tested filters for all tested scenarios. Parameters of all filters were optimized for the best performance.

	MAE [Deg]			RMSE [Deg]		
Filter Type	Scenario 1	Scenario 2	Scenario 3	Scenario 1	Scenario 2	Scenario 3
Madgwick	2.66	4.63	3.90	3.48	6.61	5.91
Valenti	2.84	5.91	4.51	3.82	7.08	6.39
Suh	5.29	10.68	11.54	6.53	15.01	15.51
Guo	2.57	4.35	3.94	3.49	6.41	5.82
SCF (ours)	**1.97**	**4.06**	**3.37**	**3.01**	**6.24**	**5.57**
FSCF (ours)	**1.97**	**4.06**	**3.37**	**3,01**	**6.24**	**5.57**

**Table 3 sensors-20-03824-t003:** Accuracy of the tested methods on testing data set. Parameters of all methods were trained either on the slow or fast rotation part of the data set. Reported accuracy is measured on the testing data, which is the remaining part of the dataset.

	MAE [Deg]		RMSE [Deg]	
Filter Type	Trained on Fast	Trained on Slow	Trained on Fast	Trained on Slow
Madgwick	3.30	5.62	4.29	7.76
Valenti	4.84	13.93	5.85	15.52
Suh	24.10	34.11	42.29	48.18
Guo	3.37	6.72	4.13	8.52
SCF (ours)	**3.20**	**5.29**	**3.91**	**7.51**
FSCF (ours)	**3.20**	**5.29**	**3.91**	**7.51**

**Table 4 sensors-20-03824-t004:** The computational cost for each filter in terms of floating-point operations.

Filter Type	Filter FLOPs
Madgwick	287
Valenti	244
Suh	352
Guo	≈1000
SCF	179
FSCF	160

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
