# Peer review of "Fast AHRS Filter for Accelerometer, Magnetometer, and Gyroscope Combination with Separated Sensor Corrections"

_sensors, 2020, doi:10.3390/s20143824_

Round 1

Reviewer 1 Report

1. General information

The content of the manuscript is interesting. It is one of the best manuscripts I have reviewed. Being not a specialist in adaptive filters, I have no concerns about the essential content of the text. However, the following issues, of a more general character, must be addressed in order to improve the overall presentation of the obtained results:

- the Abstract and Conclusions contain no numerical values related to the obtained results (I suggest to provide in both cases a synthetic summary of the content in lines 272-278; 269-271)
- it is not always clear how the authors determined the reported errors (numbers and graphs), e.g. in line 287,
- metrological parameters of the applied platform are not reported
- the bibliography must be expanded

Many lingual errors can be found throughout the text, mostly related to plural/singular ending, e.g. line 57, 64, 201.

2. The most crucial issues to be improved

- my main concern is about the reported and the existing errors; how were the percentage values reported throughout the text calculated? what was the reference for errors in Fig. 5, 6? The authors mention "desynchronization between the reference and measured data" in line 231-232 (a similar idea is introduced in line 247-248); does it mean that the real parameters describing motion of the platform were not used as the reference?
- I suggest to provide values not only in terms of MAE but RMSE as well,
- what are the metrological parameters of the applied platform? What accuracy (both under static and dynamic conditions) does it feature?; a respective datasheet should be provided as a reference; actually, more about this issue is said in the Abstract than in the body of the text,
- what is the data illustrated in Fig 4? The preset parameters controlling motion of the platform? If so, how reliable they are in the sense of errors between the set parameters and the real ones?
- what angle is presented in Fig. 5-9? an arbitrary angle? Any of the Euler angles?
- the statement in line 39 ("the measurement can not be used as a reference") is not absolutely true, since accelerometers operate correctly under static and quasi-static conditions (see e.g. DOI:10.1109/JSEN.2006.881433)
- line 41: it should be added "and various magnetic disturbances" - e.g. existing electric fields, as also stated in lines 117-118,
- Such crucial issues (as stated e.g. in DOI: 10.1007/s12541-014-0558-8) related to MEMS sensors (mainly accelerometers and gyros) as: long-term and temperature drifts, cross-axis sensitivity, misalignments of the sensitive axes, amplitude and phase attenuation over frequency are totally disregarded in the text; some relevant comments must be provided in the text,

3. Detailed remarks:

- abbreviations used in the Abstract (AHRS, SO(3), MEMS) should be defined in this section, since the Abstract should be a self-contained part of the paper,
- Fig. 1: the characters are a bit too small; a poor contrast makes them even less clear and intelligible
- Fig. 1 and the related description: would it be possible to distinguish what is corrected by what (e.g. magnetometer outputs by accelerometer outputs, not vice versa - Eq (17))?
- Fig. 3 should have some labels over it, presenting the basic elements of the system: the platform, the tested sensor (IMU); besides, the picture is too big - only the platform should be illustrated;
- in order to part the three different scenarios, I propose to add two vertical lines on the relevant graphs (Fig. 4, 9)
- what is "w" in Fig. 4, legend assigned to the reference quaternion?
- the authors refer to the used IMU by Bosch (line 188) as a "sensor", and then the refer to many sensors (e.g. in line 222), what causes a confusion,
- Tab. 2 - I suppose the correct values are 5.189; 3.368; 3.367 (the same refers to Tab, 3); I suggest to introduce a reasonable rounding of these values corresponding the accuracy of operation of the experimental platform; besides, I suggest to provide also values of RMSE values;
- line 238-239: maybe a lower value of RMSE and not MAE would be a better idea?
- Tab. 3: neither the column names nor the related description clearly states that the listed values are errors
- I suggest to stick to the same names; so it would better to use "FSCF" in the legends of Fig. 7-9 instead of "Proposed",
- what was the measure of the "best overall performance" referred to in Fig. 7-9?
- datasheet for VICON platform should be provided in the references, and its accuracy reported in the manuscript
- Fig. 1: add to the figure caption that "aRef" and "mRef" "qDot" etc., are explained in Sec. 4.1, 4.3. and 4.4 etc.; "w.." are weights
- each of the Euler angles in eq. (9) should be named
- line 114: I suggest to replace 'z' with z in italics
- could you explain what you mean by "the insecurity with implementation" in line 176?

4. Some (not all) lingual corrections

- l. 254: "detain" -> " detail"
- l. 263: "slower" -> "lower"
- l. 285: "The tests unfortunately shows the sad reality" -> "The tests proved"
- l. 286: " has in general " -> "have in general "
- l. 4: "accelerometer and magnetometer sensors" -> "acceleration and magnetic field sensors"
- l. 273: "against the second" -> "with respect to the second"
- l. 17: " in combination of magnetometers " -> " in combination with magnetometers"
- l. 18: "combination take " -> " combination takes"
- l. 37-38: "relative changes of orientation and is subject to slow drift " -> " relative changes of orientation that are subjected to slow drifts"
- l. 39: "the measurement is can not be used " -> " the measurement cannot be used"
- l. 40&41: " contaminated " -> " disturbed "
- l. 48: " theory and which assumes " -> " theory, which assumes "
- l. 83: " like " -> " as "
- l. 93: " some " -> " same "
- l. 122: " measurements mt " -> " measurements of mt "
- l. 134: " Madwick " -> " Madgwick "
- l. 163: " after 30 degrees " -> " over 30 degrees"
- l. 238: " best " -> " lowest" or " smallest "

5. References:

- over 56% are more than 10 years old, but is OK, since the authors provide a related historical outline,
- there are no references published in the journal of Sensors in 2019 or 2018 (to build the upcoming journal Impact Factor and CiteScore)
- authors do not cite any of their own works; why?
- majority of the references (11) are quoted only in the Introduction,
- 16 items are rather not a satisfactory number; add some references.

Reviewer 2 Report

The attitudes estimation from the data of the accelerometers, gyroscopes and magnetometers has been well studied in the past decades. This paper employs the basic prediction-correction structure, where the attitudes are predicted by the gyro data and then updated based on the accelerometer and magnetometer data. In particular, the authors design a new correction step that put hard limit on amplitude of the disturbance of individual sensors, which can be considered as their contribution.

The paper is technically sound and written with sufficient clarity. However, while the work reflects a good expertise and understanding of attitude and heading estimations for engineering applications, it is poor in terms of bring in any new fundamentals and/or scientific novelties for journal publication.

Reviewer 3 Report

The authors propose a new filter structure that has low computational cost and shows promising performance in accuracy. The proposed method applies the correction function to both the accelerometer and magnetometer measurements before their incorporation in sensor fusion. The comparative analyses highlighted some significant benefits of the filter over the state of art techniques.

The article has theoretical and practical significance. However, it should be corrected in many places: 1) it contains many language typos and styling issues, 2) the proposed material is not presented in sufficient detail in some places which results in that the results cannot be repeated based on the current form of the paper.

English needs to be improved. Extensive editing is required (both language and style)

Both filters are quaternion-based without any redundant steps. -> Both approaches are quaternion-based filter realizations without redundant steps.

The MARG combination take advantage-> The MARG combination takes advantage

The aim AHRS system -> The aim of the AHRS system…

Many followup contributions focused on efficient solution of this problem -> Many follow-up contributions focused on the efficient….

acclerometer -> accelerometer

stochastic filtering theory and which assumes that the -> stochastic filtering theory and assumes that the…

distribution of the state evolution error and observation errors -> distribution of both the state evolution and observation errors.

While the Kalman filtering is a -> While the Kalman filter is a

based on assumption that -> based on the assumption that

we follow the approach of [14] -> we follow the approach presented in [14]

disturbance affects direction of the correction step -> disturbance affects the direction of the correction step

we have to replace the Wahba problem by an alternative. -> we have to replace the Wahba problem by an alternative approach.

new features will be closely described in next chapter. -> new features are described in the next chapter.

In this Section -> in this section

First, rotation of normalized vectors in Euclidean space -> First, rotation of normalized vectors in Euclidean space is presented

then elementary operations with quaternions. -> then elementary operations with quaternions are described.

There were the some attempts to do -> There were some attempts to do

separation of sensor contribution -> separation of sensor contributions

The proposed filter is rotational -> ?

from Fig. 1 (right) will be now described. -> from Fig. 1 (right) is described as follows.

The first step is transformation -> The first step is the transformation

to the Euler angles -> to Euler angles

The second step is transformation -> The second step is the transformation

Madwick -> Madgwick

Linear Madgewick -> Linear Madgwick

(5.3.1) -> section (5.3.1)

of Fig. 1. -> in Fig. 1.

experimental part, Section 5.3.2. -> experimental part, see section 5.3.2.

Tab. 2 -> Table 2

Tab. 3. -> Table 3.

analyzed in detain Fig. 9 -> analyzed in detail in Fig. 9.

Abstract

Abbreviations are required be introduced first. Then those can be used throughout the paper, i.e., attitude and heading reference system (AHRS). Too many keywords are supplied.

Introduction

Literature overview

The introduction does not highlight the recent advances of attitude techniques. Most of the cited papers are quite old. The authors describe the deterministic approaches from 2000, but does not mention or analyze the advantages of most recent solutions in the topic, such as:

Wu, J.; Shan, S. Dot Product Equality Constrained Attitude Determination from Two Vector Observations: Theory and Astronautical Applications. Aerospace 2019,

Liu, F.; Li, J.;Wang, H.; Liu, C. An improved quaternion Gauss–Newton algorithm for attitude determination using magnetometer and accelerometer. Chin. J. Aeronaut. 2014

Wu, J.; Zhou, Z.; Fourati, H.; Cheng, Y. A super fast attitude determination algorithm for consumer-level accelerometer and magnetometer. IEEE Trans. Consum. Electron

Similarly, the authors describe the Kalman filter as a suboptimal estimator, however only three papers are mentioned (one from 2006 and two from 2017). It is highly required to update the literature overview of most recent techniques in the realm of Kalman filtration: what are the most recent techniques, what are the advantages and issues presented in the recent papers.

The authors do not discuss the adaptive strategies, which actually provide the most effective way to handle time varying disturbances resulting from motion or ferromagnetic materials. The authors are required to extend the introduction with the description of adaptive solutions: what are the recent approaches, what are the effective ways of handling disturbances, what are the issues and so on.

The authors address the paper of Madgwick’s approach from 2011. However, that approach has been significantly improved recently:

Wilson, S.; Eberle, H.; Hayashi, Y.; Madgwick, S.O.; McGregor, A.; Jing, X.; Vaidyanathan, R. Formulation of a new gradient descent MARG orientation algorithm: Case study on robot teleoperation. Mech. Syst. Signal Process. 2019

Why don’t discuss the authors the differences between the improved filter and their solution?

Contribution

The contribution of the paper is not described in the introduction.

Reminder of the paper is missing

The paper structure is not described, such as: what is discussed in section 2, what is the aim of section 3, etc..

Section 2

It seems like section 2 describes some sort of contribution of the paper and the main results. I recommend moving section 2 to the introduction under a subsection called Contribution of the paper. Since the contribution is crucial to be clear I recommend emphasizing both the results and findings of the paper. The contribution should be clear.

Section 3

The concept of unit vector, direct angle and cross product is quite elementary, I do not think, that these should be introduced.

Equations 3 and 4 make redundancy. One of those equations is enough.

Section 4

The authors state that high sampling rate enables the linearization of trigonometric function. This statement is a bit strong and needs further proof at least. What is the sampling rate? What is the dynamics of the system to be captured?  The authors mention 100 Hz in section 5.3, which is a quite slow data acquisition speed, and at that speed the error resulting in from linearization can be quite significant.

“The impact of this approximation is negligible in real conditions.” – has this statement been proven in real conditions?

Eq. (17) needs to be explained in more detail. The standard method here is to use the estimated quaternion to obtain the magnetic reference vector, however the authors use the predicted acceleration in body frame instead, and state that it “has negligible difference from the original formulation”. Please give more detailed description.

In Eq. (20),  theta_a is not introduced.

Section 5

“Tuning parameters of each method were optimized using brute force search.”

“To obtain fair testing conditions, parameters of all tested filters were optimized for each test and each filter to get the best possible response.”

“Parameters of all filters were tuned for best overall performance.”

Please describe the aforementioned process in detail.

This needs to be described in detail. How did the authors set the parameters of each filter? Did the authors use optimization algorithm? How did the authors grant fair comparison? How did the authors achieve the best performance of each filter? Since the correct choice of filter parameters influence the most the filter performances this description is very important.

A table with the selected filter parameters is required in section 5.

The authors did not write about the embedded system which executes the algorithm. Which microcontroller/processor was used in the experiment? Maybe a block diagram would have been beneficial of the system.

I assume Table 3 contains the MSE results, however it is not clear. Please indicate in the table.

I would be good to see on a Figure the filter outputs (not just the error). For example, the filter outputs in case of Figure 9.

Based on three simple scenarios (sum time of the experiment was 90 sec), and without the comparison with new the techniques, it is too strong to state: "sad reality that many new filters presented in this field has in general no better behavior than the old ones".

Round 2

Reviewer 2 Report

I have no further questions.

Author Response

Thank you once again for your care and attention to our manuscript.

Reviewer 3 Report

The authors provided the updated version of the manuscript. They considered all my comments and extended their manuscript accordingly. I appreciate these extensions and I found the current form more appropriate for publication.

However, there are still three main concerns that needs to be addressed very carefully/consciously by the authors.

  • The authors corrected all the language issues I recommended. However, I am not a native English speaker and my suggestions were just some examples I found when I read the article.

The article still requires English correction via a professional service, such as the MDPI English Editing service or other professional service. Without professional English correction I think the article does not sound enough technical and contains many offhand sentences. There are too many sentences that are not well structured and make the understanding difficult.

  • The sections have been updated to acceptable level as it was recommended in the comments. The introduction has been updated with the state of art solutions. Additionally, each section has been updated to make the deductions clearer. I appreciate these changes. However:

Use citation for Eq. (7), what is the source that equation?

Eq (16) still seems unclear. The authors use the estimated acceleration in dot product with the current magnetometer measurement. This approach is unclear for me. Please provide a more detailed description of this approach (even a figure can be beneficial to make the understanding clearer) or use citation to the literature that use this method.

  • My main concern is related to the experimental validation section. I see that the authors optimized the parameters for each scenario separately. I don’t really agree with this approach. This tells me, that based on the data set (scenario) different parameters should be used to provide good performance. Which is fine if the filter has adaptation laws. However, the proposed filter is a fixed gain filter, therefore the gains should have been fixed for each experiment (for each filter).

There should be training data on which the tuning of filters is performed and there should be experimental data on which the performance is evaluated.

Table A1 tells me that the data sequence(s) used for final evaluation and comparison to the other filters were not different from the ones applied for optimization. These should be different to show that the obtained solution is general enough, not overfitted to one data sequence.
